# Variation of Female Pronucleus Reveals Oocyte or Embryo Chromosomal Copy Number Variations

*Jingwei Yang, Yikang Wang, Chong Li, Wei Han, Weiwei Liu, Shun Xiong, Qi Zhang, Keya Tong, Guoning Huang,\* and Xiaodong Zhang\**

The characteristics of the human pronuclei (PNs), which exist 16–22 h after fertilization, appear to serve as good indicators to evaluate the quality of human oocyte and embryo, and may reflect the status of female and male chromosome composition. Here, a quantitative PN measurement method that is generated by applying expert experience combined with deep learning from large annotated datasets is reported. After mathematic reconstruction of PNs, significant differences are obtained in chromosome-normal rate and chromosomal small errors such as copy number variants by comparing the size of the reconstructive female PN. After integrating the whole procedure of PN dynamics and adjusting for errors that occur during PN identification, the results are robust. Notably, all positive prediction results are obtained from the female propositus population. Thus, the size of female PNs may mirror the internal quality of the chromosomal integrity of the oocyte. Embryos that develop from zygotes with larger female PNs may have a reduced risk of copy number variations.

J. Yang, C. Li, Q. Zhang, G. Huang, X. Zhang
Center for Reproductive Medicine
Women and Children's Hospital of Chongqing Medical University
Chongqing Health Center for Women and Children
Chongqing 400010, China
E-mail: gnhuang217@sina.com; zhangxd207@163.com
J. Yang, C. Li, W. Han, W. Liu, S. Xiong, Q. Zhang, K. Tong, G. Huang, X. Zhang
Chongqing Key Laboratory of Human embryo Engineering
Chongqing 400010, China
Y. Wang
Department of Mechatronics
Graduate School of Medicine, Engineering, and Agricultural Sciences
University of Yamanashi
Yamanashi-ken 400-8510, Japan
W. Han, W. Liu, S. Xiong, K. Tong, G. Huang, X. Zhang
Chongqing Clinical Research Center for Reproductive Medicine
Chongqing 400010, China

## 1. Introduction

Human embryos begin with the fertilization of an oocyte by a sperm. A sperm binds to the oocyte and delivers its chromosomal DNA, triggering the beginning of a series of microscopic events in the zygote including the cortical granule reaction, which prevents poly-fertilization, expelling of the second polar body, and the formation and migration of two separate pronuclei (PNs) that carry the maternal and paternal chromosomes. The male and female PNs form in proximity to the zygote's surface. Then, they need to move inward in order to unite the paternal and maternal chromosomes on the first mitotic spindle.[1] During PN migration from the periphery inward to the center of the zygote, the areas of both PNs increase gradually.

Phenomena related to pronuclear and nucleolar movements were first described by Wright et al.[2] Notions including pronuclear alignment and uneven/even numbers of chromosomes in the PN and nucleolus precursor bodies (NPBs) have been expressed in more distinct pronuclear scores and used as a means to select embryos based on the Z-score.[3] The scores have been correlated with improved embryo development,[4,5] increased pregnancy and implantation rate,[6–8] and embryonic chromosomal content[9–11] after the pre-implantation genetic test (PGT). However, some studies have disputed the effect of pronuclear scores for in vitro fertilization (IVF) or intracytoplasmic sperm injection (ICSI),[12–14] even if 0PN- and 1PN-derived blastocysts have similar neonatal results as 2PN-derived blastocysts.[15,16]

In theory, characteristics of the pronuclear stage may mirror the internal chromosomal integrity of the oocyte and the sperm.[11,17,18] Meanwhile, developmental details such as disorder cleavage, embryonic fragment extrusion, uneven blastomeres, and abnormal morphokinetics during the post-zygote stage (cleavage, morula, and blastocyst stage) might reflect embryonic developmental dysfunction, mainly aneuploidy, and mosaicism.[11–22] Due to the vague standard of methods (more than 6 scoring systems) in current pronuclear assessments,[12] the effect of pronuclear scores remains unclear. The dynamic character of the PN, incongruent practice in IVF laboratories such as fertilization time and checking time, and the heterogeneity in patients make efficient qualitative classification for pronuclear assessment impossible.

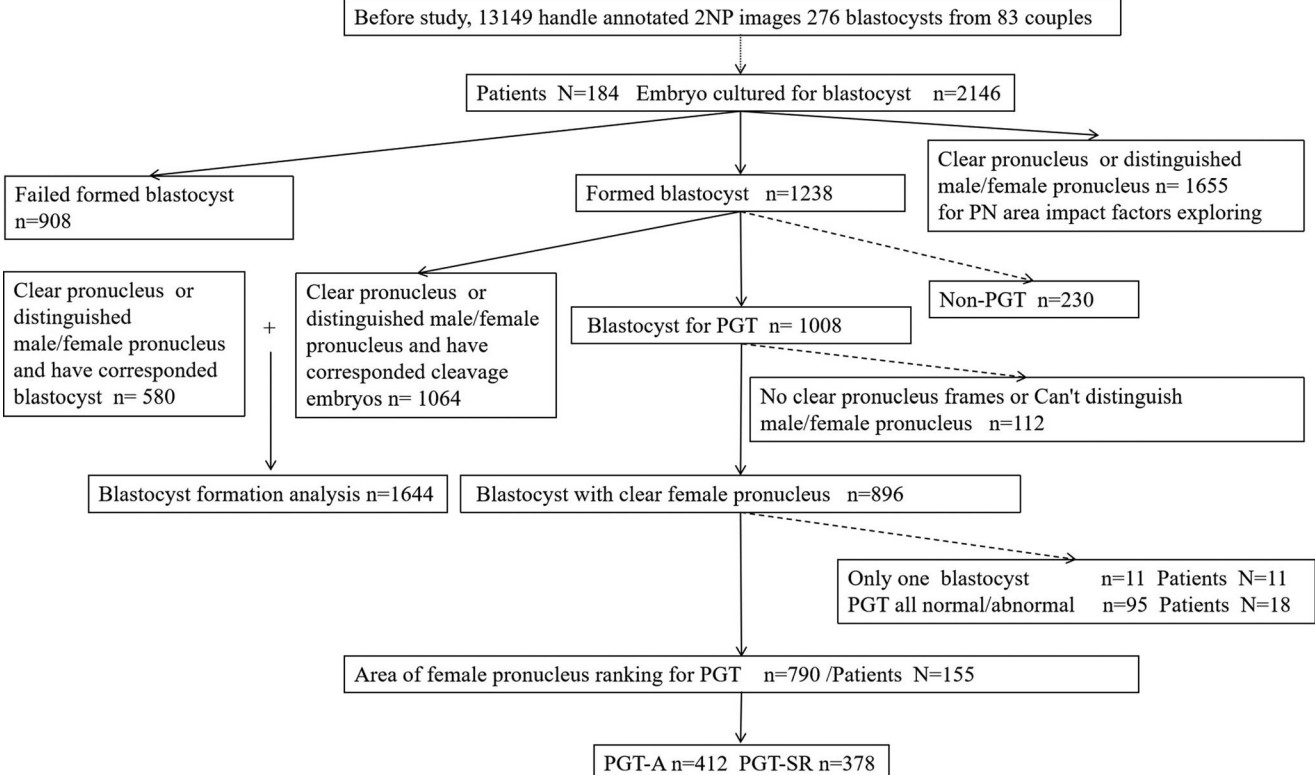

**Figure 1.** Flow chart.

Here, we aim to construct a computer-assisted algorithm for quantitative analysis for pronuclear assessment in ICSI patients from time-lapse incubators and test its efficacy in the diagnosis of chromosomal integrity in oocytes or embryos.

## 2. Results

Before the study, 13 149 handle-annotated 2NP images of 276 blastocysts from 83 couples were used for Mask R-CNN learning(R-CNN) learning to PN recognition and establishment of the algorithm for quantitative analysis for pronuclear assessment.

In this study, 2146 embryos from 184 patients were cultured for blastocysts and 1238 of them formed blastocysts. Blastocyst-stage embryos with unclear PN frames, indistinguishable male/female PNs, or only one blastocyst, as well as all normal/abnormal PGTs, were excluded from the final analysis. In total, 155 couples with 790 blastocyst-stage embryos were included in the final analysis, with 412 and 378 embryos in the pre-implantation genetic test for aneuploidy (PGT-A) and pre-implantation genetic test for chromosome structural rearrangements (PGT-SR) subgroups, respectively (**Figure 1**).

After Mask R-CNN learning of 13 419 images morphometrically labeled by two experienced embryologists defined as handle-annotated 2PN images (276 embryos from 83 couples), the number of frames for artificial intelligence (AI) automatic recognition reached 16 634 for this study. After comparison with the handle annotation database using 16 634 AI automatic recognized images, it was found that 3343 more images were misla-

beled than the actual number of 2PN images (overestimated). In these 3343 images, 3192 images (95.48%) came from early-stage PNs (12 h post-insemination), the rest came from 12–14 h post-insemination and no images came from 14 h post-insemination. Additionally, 128 images were missing because of partial overlap of two PNs in middle- or late-stage PNs (14–22 h post-insemination) (**Figure 2**A).

Figure 2 Female PN reveals oocyte or embryo abnormality.

The accuracy of distinguish PN numbers by Mask R-CNN learning for (0, 1, 2, 3) reached 80.06% [13 419/(13 419 + 3342 + 128)] in recognition of all PN stages, 97.9% [13 419/(13 419 + 3342 − 3192 + 128)] in recognition at 12 h post-insemination until PN disappearance, and 99.06% [13 419/(13 419 + 128)] at 14 h until PN disappearance. No error was found in AI boundary drawing of PNs after handle checking except for PN number recognition-related boundary errors (e.g., mismarking vacuoles as PNs). For above errors in female and male PN coefficient $\beta 1$ calculation, original, adjusted (first and last three images deleted due to the high inaccuracy rate in recognition and high weight in the fitting curve model), 12 h post-insemination, and 14 h post-insemination values were extracted for effective testing. Then, 2146 embryos from 184 patients who have top-quality blastocysts for PGT were included in the data analysis. The baseline characteristics of patients and their IVF outcomes are shown in **Table 1**.

Different grades of PN identification are shown in Figure 2B. In total, 529 from 2146 zygotes (24.65%) underwent handle male/female PN reversal after computer marking and embryologist checking. Total frames of clear and distinguished 1655 2PN embryos reached 108 587 images and these images were included

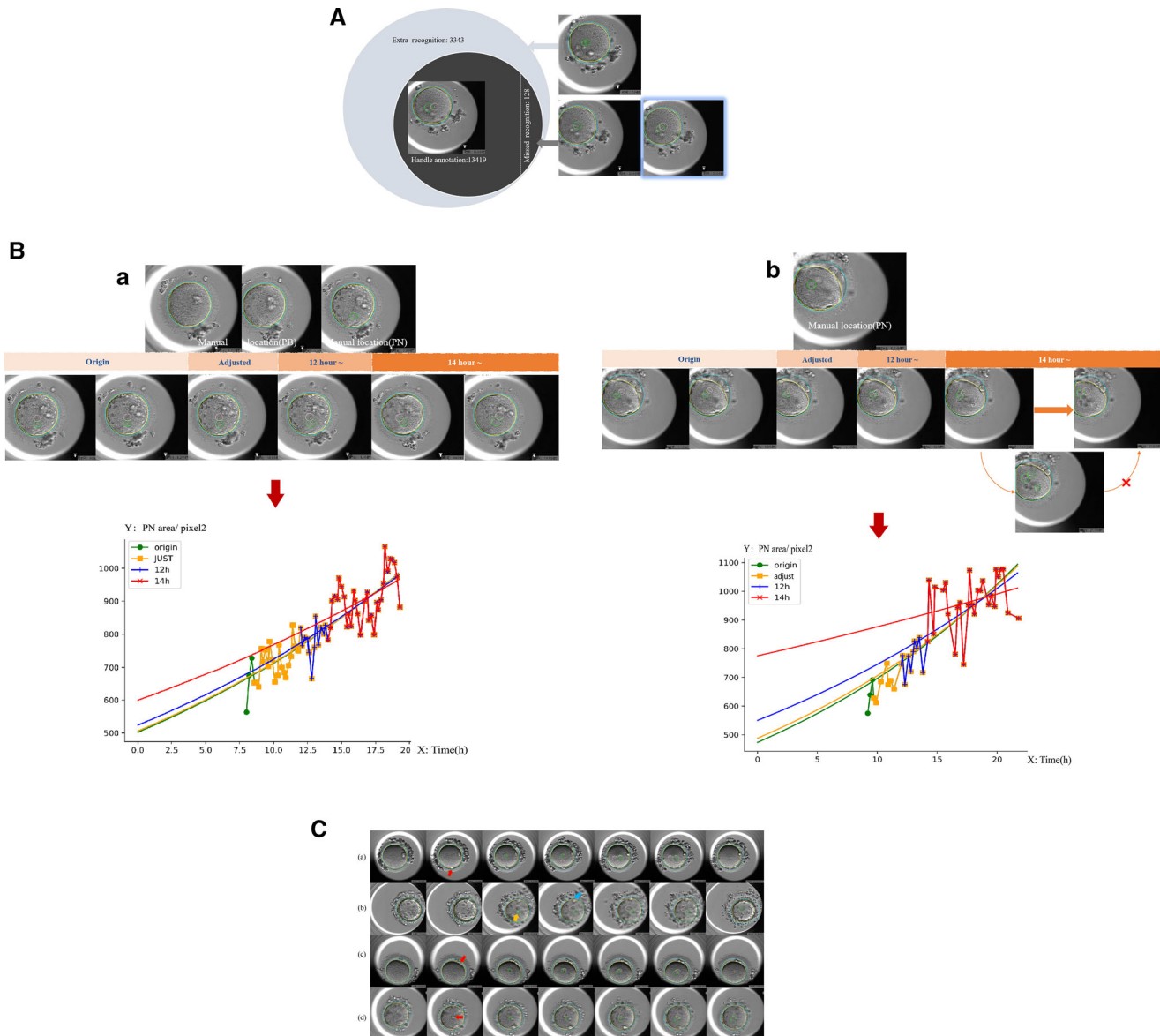

**Figure 2.** PN developmental model and adjustment of coefficients. A total of 16 634 images from 276 embryos of 83 couples. A) The frames of AI automatic recognition and handle annotation. B) Assessment of pronucleus (PN) coefficients. a) All automatically recorded PN areas (the scatters in the figure) were included in the original fitting curve (green line). The lowest value was the original coefficient $\beta 1$. The outlier PN areas of the first three frames (which have the highest variance) were deleted in the adjusted fitting curve (yellow line). 12 h post-insemination and 14 h post-insemination fitting curves (blue and red lines) are depicted based on the lower automatic PN recognition error rates in those stages, which makes it easier to distinguish based on the value of $\beta 1$. b) Sometimes, errors in counting occurred in automatic PN recording, but all data with incorrect numbers of PNs (other than pink and green PN circles), such as 3PN (mislabeling a vacuole as a PN), were omitted in fitting curves. C) The identification of polar bodies (PBs) and female/male PN handle checking. Red arrow, the appearance of the second PB; yellow arrow, the appearance of the male PN; blue arrow, dubitable second PB. a) Easily distinguished PN frames, clear PB2, and suitable distances of female and male PN to PB2 (proportion: 75%). b) Difficult to distinguish PN frames due to the cumulus cell disturbance (proportion: 6%). c) Difficult to distinguish PN frames due to the overlap of female and male PNs (proportion: 5%). d) Difficult to distinguish PN frames due to the equidistance of female and male PNs to PB2 (proportion: 14%).

to explore the factors that impacted female and male PN areas. In total, 1644 embryos with 108 028 images were included for blastocyst formation analysis. Finally, 790 embryos with 52 479 images from 155 patients were included for the analysis of both areas of PNs and PGT results.

No clinical or cell biological factors were correlated with the female PN coefficient $\beta 1$ except for the male PN coefficient $\beta 1$

($r = 0.75$, $P < 0.01$, **Table 2**) and the distribution data of the pronuclear area showed significant heterogeneity in individual patients. This heterogeneity makes it impossible to find a normal range of coefficient $\beta 1$ (original data: $Q = 96.32$, $df = 183$, $I^2 = 95.6\%$, Figure S1, Supporting Information). For homogenized exploration of the correlation between PN area coefficient $\beta 1$ and PGT results, ranking orders (biggest to smallest, Figure

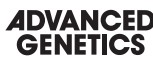

**Table 1.** Baseline characteristics of patients.

| Patient characteristics | Mean (SD)/percent |
|---|---|
| Age at IVF cycle | 30.55 ± 3.94 |
| Retrieved oocytes | 16.79 ± 6.81 |
| AMH level | 4.30 ± 2.77 |
| FSH (basic) | 5.59 ± 2.60 |
| BMI | 21.77 ± 2.76 |
| Gn day | 9.26 ± 1.56 |
| Gn dose | 1875.08 ± 622.27 |
| MII oocyte | 14.09 ± 6.20 |
| Infertility diagnosis (%) | |
| Decreased ovarian reserve | 2.17 (4/184) |
| Habitual loss | 9.23 (17/184) |
| Male chromosome abnormality | 35.87 (66/184) |
| Female chromosome abnormality | 32.07 (59/184) |
| Uterine/fallopian tube factor | 7.61 (14/184) |
| Obstetric abnormality | 5.43 (10/184) |
| Unexplained | 7.62 (14/184) |
| IVF protocol (%) | |
| Gn-a | 42.93 (79/184) |
| Gn-ant | 57.07 (105/184) |

**Table 2.** Correlation between male and female pronuclear coefficient $\beta 1$.

| Dependent variable: Female pronuclear (coefficient $\beta 1$) | Standard $\beta$ | P |
|---|---|---|
| Male pronuclear | 0.75 | 0.000 |
| Gn day | −0.08 | 0.092 |
| Gn dose | 0.08 | 0.051 |
| AMH | 0.02 | 0.409 |
| BMI | 0.07 | 0.200 |
| FSH (basic) | −0.01 | 0.755 |
| Female age | −0.04 | 0.095 |
| MII oocytes number | 0.04 | 0.232 |
| Viable embryo number | −0.06 | 0.064 |
| Infertility diagnosis | −0.02 | 0.322 |
| IVF protocol | 0.02 | 0.496 |

S2, Supporting Information) of male/female were employed for testing in every patient. In blastocyst formation analysis, 1064 zygotes successfully developed into blastocysts and 580 failed, and no significant relationship was observed between ranking order of the female or male PN coefficient $\beta 1$ and blastocyst formation (**Table 3**).

PGT results from transferable blastocysts have a U curve in the female age-dependent distribution (Figure S3, Supporting Information). By the female PN original coefficient $\beta 1$ ranking, chromosome normal rate ("chromosome-normal" and "sole mosaic" proportion in total embryos) in the blastocyst with biggest PN area (top 1) is much higher than that of the blastocyst with smallest PN (last 1) (58.06% versus 45.16%, OR = 1.68 [1.07–2.64], $P = 0.031$), and the chromosome-normal rate of the top 2 blastocysts is higher than that of the last 2 blastocysts but with

no statistical difference (59.31% versus 50.49%, $P = 0.091$) (Table S1-1, Supporting Information). After adjusting for coefficient $\beta 1$, the chromosome-normal rate in the top 1 blastocyst is much higher than that of the last blastocyst (58.71% versus 45.16%, OR = 1.73 [1.10–2.71], $P = 0.023$), and the chromosome-normal rate of the top 2 blastocysts is higher than that of the last 2 blastocysts but with no statistical difference (57.35% versus 50.00%, $P = 0.164$) (Table S1-2, Supporting Information). For coefficient $\beta 1$ 12 h post-insemination, the chromosome-normal rate in the top blastocyst is much higher than that of the last blastocyst (64.52% versus 43.23%, OR = 2.39 [1.51–3.77], $P < 0.001$], and the chromosome-normal rate in the top 2 blastocysts is higher than that of the last 2 blastocysts (63.73% versus 47.55%, OR = 1.94 [1.30–2.88], $P = 0.001$) (Table S1-3, Supporting Information). For coefficient $\beta 1$ 14 h post-insemination, the chromosome-normal rate in the ranking top blastocyst is much higher than that of the last blastocyst (66.45% versus 42.58%, OR = 2.61 [1.68–4.24], $P < 0.001$), and the chromosome-normal rate in the top 2 blastocysts is higher than that of the last 2 blastocysts (64.22% versus 48.04%, OR = 1.94 [1.31–2.89], $P = 0.001$) (Table S1-4, Supporting Information). The trend that the top blastocysts showed higher chromosome-normal rates can be observed in **Table 4** and **Figure 3**A. However, for the male PN coefficient $\beta 1$, no significant difference was observed (Figure 3B).

In all patients without distinguishing PGT-A and PGT-SR, the first and last ranking order could be used to detect the chromosome-normal and sole mosaic embryos (46.45% versus 28.39% in chromosome-normal rate, 66.45% versus 42.58% in chromosome-normal plus mosaic rate) (**Figure 4**A, Table S2, Supporting Information). No significant differences in chromosomal status were obtained by coefficient $\beta 1$ ranking in the PGT-A population (Figure 4B; Tables S4 and S5, Supporting Information). However, in the PGT-SR population, the first order embryos in coefficient $\beta 1$ ranking have higher chromosome-normal and sole mosaic rates than the last embryos (40.54% versus 21.62% in chromosome-normal rate, 66.22% versus 32.43% in chromosome-normal plus mosaic rate) (Figure 4C, Table S6, Supporting Information). Relatively smaller chromosomal errors defined as "euploid with errors" and "sole deletion and/or duplication" were significantly different between first-order and last-order embryos, even with median-order embryos having a hierarchical difference (PGT 9.68% versus 15.41% versus 20%, PGT-SR 13.51% versus 26.92% versus 36.49%, respectively, $P < 0.001$) (Figure 4A,C, Tables S3, S6, and S7, Supporting Information). For population- and chromosomal error-stratified analysis of coefficient $\beta 1$ at 14 h post-insemination, PGT-A and PGT-SR, "aneuploidy with errors" and "euploidy with errors" (and a more detailed error classification including "chromosome-normal," "sole mosaic," "sole aneuploidy," and mosaic forms, "sole deletion and/or duplication" and mosaic forms, and "complex chromosomal errors"), "embryo's chromosomal error coincident/inconsistent with female," and "embryo's chromosomal error coincident/inconsistent with male" are shown in Figure 4B,C (Tables S4–S9, Supporting Information). From the female propositus and embryo analysis in PGT-SR, coefficient $\beta 1$ ranking has detection power in both coincident and inconsistent chromosomal errors (28.57% versus 50%, 9.52% versus 26.19%, $P < 0.05$, respectively; Figure 4C, Table S8, Supporting Information), which implied inherited and novel errors in

**Table 3.** Chromosome-normal rate between blastocysts and embryos that failed to form blastocysts based on the rank of the pronucleus area coefficient $\beta 1$.

| Rank | Female | | Male | |
|---|---|---|---|---|
| | Formed blastocyst | Failed | Formed blastocyst | Failed |
| 1 | 126/211 (59.7%) | 85/211 (40.3%) | 145 (63.04%) | 85 (36.96%) |
| 2 | 130/211 (61.6%) | 81/211 (38.4%) | 117 (63.59%) | 67 (36.41%) |
| 3 | 125/199 (62.8%) | 74/199 (37.2%) | 127 (61.65%) | 79 (38.35%) |
| 4 | 124/178 (69.7%) | 54/178 (30.3%) | 128 (63.05%) | 75 (36.95%) |
| 5 | 108/167 (64.7%) | 59/167 (35.3%) | 105 (61.4%) | 66 (38.6%) |
| 6 | 95/137 (69.3%) | 42/137 (30.7%) | 91 (67.41%) | 44 (32.59%) |
| 7 | 78/115 (67.8%) | 37/115 (32.2%) | 76 (71.03%) | 31 (28.97%) |
| 8 | 52/92 (56.5%) | 40/92 (43.5%) | 63 (71.59%) | 25 (28.41%) |
| 9 | 55/75 (73.3%) | 20/75 (26.7%) | 54 (72.97%) | 20 (27.03%) |
| 10 | 41/59 (69.5%) | 18/59 (30.5%) | 52 (73.24%) | 19 (26.76%) |
| >10 | 130/200 (65.0%) | 70/200 (35.0%) | 106 (60.57%) | 69 (39.43%) |
| Total | 1064 | 580 | 1064 | 580 |
| Pearson Chi-Square | 12.94 | | 7.18 | |
| P | 0.227 | | 0.712 | |

**Table 4.** Rate of chromosome-normal blastocysts by the female PN coefficient $\beta 1$.

| Rank | Origin | Adjusted | 12 h | 14 h |
|---|---|---|---|---|
| 1 | 90/155 (58.06) | 90/155 (58.06) | 99/155 (63.87) | 102/155 (65.81) |
| 2 | 79/155 (50.97) | 75/155 (48.38) | 82/155 (52.9) | 79/155 (50.97) |
| 3 | 78/138 (56.52) | 82/138 (59.42) | 69/138 (50) | 70/138 (50.72) |
| 4 | 46/107 (42.99) | 47/107 (43.93) | 50/107 (46.73) | 49/107 (45.79) |
| 5 | 40/81 (49.38) | 39/81 (48.15) | 37/81 (45.68) | 40/81 (49.38) |
| 6 | 33/56 (58.93) | 33/56 (58.93) | 31/56 (55.36) | 29/56 (51.79) |
| 7 | 23/37 (62.16) | 23/37 (62.16) | 17/37 (45.95) | 23/37 (62.16) |
| 8 | 13/25 (52) | 14/25 (56) | 17/25 (68) | 14/25 (56) |
| >8 | 16/36 (44.44) | 15/36 (41.67) | 16/36 (44.44) | 12/36 (33.33) |

embryos, but no significant detection ability in male propositus and embryos (Figure 4C, Table S9, Supporting Information).

## 3. Discussion

A clear relationship was observed between female PNs and chromosome-normal rate in blastocyst-stage embryos for both the original and adjusted analysis, but not for male PNs. In the stratified analysis, female PNs in the PGT-SR group, but not in the PGT-A group, can be used to distinguish relatively small chromosomal errors, such as "deletion and/or duplication" and mosaic forms. Inherited and novel errors in embryos could be found using female PN ranking in female diagnosis of the PGT-SR group. The overall positive pool effect of female PN diagnosis of chromosomal errors might be caused by the PGT-SR subgroup. The negative result in PGT-A might be because of a high false-positive rate [abnormal trophectoderm (TE) but normal inner cell mass (ICM)] as well as a false-negative rate (normal TE but abnormal ICM) in this technique.[23]

From the PGT result, a high coincidence U curve has been found as previously reported,[24] but a small difference between these studies is that our age-distributed samples were blastocyst-stage embryos, not oocytes. Thus, chromosomal errors occurred post-PN from the cleavage to the morula and the blastocyst stage, and potential embryo self-correction in a later stage could reduce the power of PN predictors.[25–27]

The previously reported results indicated that embryonic chromosomal abnormality is more likely to be caused by eggs, especially in meiosis, and female PN developmental quantification could unveil the potential correlation.[27–33] Again, the results confirmed that the pronuclear stage may mirror the internal oocyte quality and chromosomal integrity. However, due to the low chromosomal error rate in sperm, male PNs had no predictive value.[34] Thus, we excluded chromosomal mosaicism because the typical mitotic errors could not be associated with the PN stage, but the effects of mitotic errors will merge in cleavage- and blastocyst-stage embryos.[35] In the earliest design of outcome measurements, the total chromosomal substance was classified as normal, deletion, and duplication, but no significant difference was obtained (Table S10, Supporting Information). Interestingly, when the outcome measure was changed into normal and abnormal, a clear difference was observed. The exact reason will be studied in further research.

A higher correlation has been obtained between female and male PN coefficient $\beta 1$, but no clinical or cell biological factor exhibited a similar correlation in subsequent analyses. The high heterogeneity of the PN coefficient $\beta 1$ made it impossible to establish a normal and abnormal range in clinical practice. No relationship between female or male PN and blastocyst formation was found, revealing that protein and energy storage could be more important to the developmental viability of embryos than chromosomal normality, at least if chromosomal errors are not too big.[36–40] The lack of NPB assessment in our model was the main drawback in predicting the occurrence of aneuploidy. From

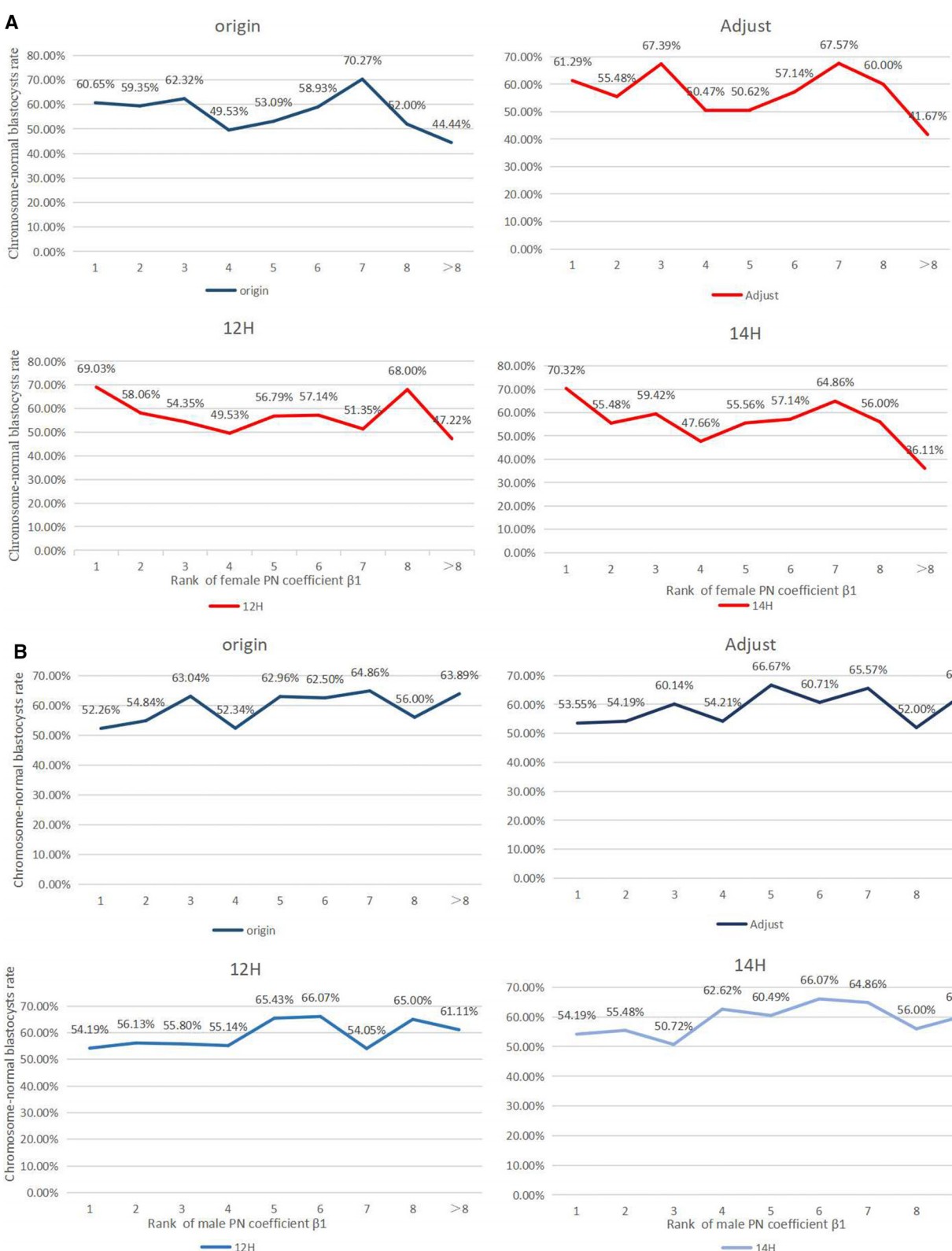

**Figure 3.** Rate of chromosome-normal blastocysts by ranking of PN coefficient $\beta1$. A) The rate of chromosome-normal blastocysts by the ranking of female PN coefficient $\beta1$. *There is correlation between ranking of female PN coefficient $\beta1$ and chromosome-normal rates. The correlation coefficients used for Spearman correlation analysis were as follows: 0.058 (Adjust), 0.099 (12 h), 0.114 (14 h), and $P$-value < 0.01 (blastocysts, $n = 790$, and patients, $n = 155$). B) The rate of chromosome-normal blastocysts by the ranking of male PN coefficient $\beta1$. *There is no correlation between ranking of male PN coefficient $\beta1$ and chromosome-normal rates (in origin/adjust/12 h/14 h). For Spearman correlation analysis, $P$-value > 0.05 (patients, $n = 155$).

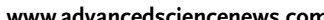

**Figure 4.** Correlation between PGT results and PNs coefficient $\beta 1$ ranking. A) PGT results (blastocysts, $n = 790$, and patients, $n = 155$). B) PGT-A results (blastocysts, $n = 378$, and patients, $n = 75$). C) PGT-SR results (blastocysts, $n = 412$, and patients, $n = 80$).

this perspective, our quantitative measurement method could assess the area of PNs and identify small chromosomal errors.

Further study is needed to improve image quality and to develop an optimal deep learning algorithm to achieve NBP recognition.

## 4. Experimental Section

The deep learning images were obtained from the ICSI cycles of 184 infertile couples requiring assisted reproductive technology (ART) therapy performed in 2019–2020 at the Reproductive and Genetic Institute of Chongqing in China. Infertility was diagnosed according to either female or male chromosomal/genomic abnormality PGT-SR, PGT-A, unexplained reason PGT-A, and tubal and pelvic factors combined with male chromosomal/genomic abnormality for ICSI and subsequent PGT-A. The study was approved by the medical ethics committee of Chongqing Health Center for Women and Children. The morphology parameters in early stage of human embryos assisted by AI were associated with genomic status (ID NO:2019-602). In total, 155 couples with 790 blastocyst-stage embryos were included in the final analysis, with 412 and 378 embryos in the PGT-A and PGT-SR subgroups, respectively (Figure 1).

### 4.1. Practices in ART

Before the ovaries were stimulated with recombinant FSH (Gonal-F, Merck Serono, Switzerland), downregulation was performed using a GnRH agonist (Decapeptyl; Ferring, Switzerland). Next, hCG (Ovidrel; Merck Serono, Italy) was administered when at least three leading follicles attained a mean diameter of >18 mm.

The flexible GnRH antagonist regimen included rFSH (Gonal-F; Serono, Aubonne, Switzerland) injection starting on day 2 of the menstrual cycle. The starting dose of rFSH was 75–300 IU daily and was customized according to the patient's age, body mass index, antral follicle count, and baseline E2, P, FSH, and LH concentrations. Cetrorelix acetate (Cetrotide; Merck Serono Ltd., Aubonne, Switzerland) was used as the GnRH antagonist. Treatment with rFSH and cetrorelix acetate was continued until the day of the final oocyte maturation trigger.

Transvaginal oocyte retrieval was performed 36 h after hCG injection. Cumulus-enclosed oocytes were collected in 2.5 mL of IVF medium (G-IVF, Vitrolife Sweden AB, Sweden) and incubated at 37 °C under 5% $O_2$ and 6% $CO_2$ conditions for insemination.

Furthermore, sperm cells with normal morphology were selected, immobilized, and then microinjected into the oocyte cytoplasm 2–4 h after oocyte retrieval. Injected oocytes were then transferred into G-1 (Vitrolife, Sweden) medium droplets and placed into microwells of a custom-made dish (EmbryoSlide, Vitrolife Sweden AB, Sweden) containing 50 µL of equilibrated G-1 (Vitrolife Sweden AB, Sweden) microdroplets over the microwells and covered with 2.5 mL of Ovoil (Vitrolife Sweden AB, Sweden). Subsequently, the dish was immediately stored in a time-lapse (TL) system (EmbryoScope, Vitrolife, Göteborg, Sweden). After 3 days of culture, the embryos were extracted and transferred to a new dish containing 50 µL of equilibrated G-2 (Vitrolife Sweden AB, Sweden) microdroplets over the microwells and covered with 2.5 mL of Ovoil (Vitrolife Sweden AB, Sweden). The TL image acquisition was set every 10–15 min at seven different focal planes for each embryo. Images (1280 × 1024 pixels) were acquired using a Leica 20 × 0.40 LWD Hoffman Modulation contrast objective specialized for 635-nm illumination.

Transferable blastocysts were defined as follows: at least in the blastocyst stage at day 5 (120 h after ICSI) with moderate expansion, having easily discernible tightly compacted inner cell mass (ICM), and having trophectoderm (TE) either in many cells forming a cohesive epithelium or in few cells forming a loose epithelium.

### 4.2. TL Setting

The data were multi-view Hoffmann modulation contrast (HMC) microscopic images of developing cells in 11 different focal segments (−75, −60, −45, −30, −15, 0, 15, 30, 45, 60, 75) taken every 15 min. HMC is a kind of oblique lighting technology commonly used in IVF.[41] When oblique light irradiates the sample, it refracts and diffracts. The light line generates different shadows through the objective lens optical density regulator so that the surface of the transparent sample produces a light and shade difference to enhance the contrast. The diameter of EmbryoSlide (Vitrolife, Switzerland) was 250 µm. Therefore, the total area of the well was 49062.5 µm.[2] The number of pixels of the well of the culture dish was measured in all the time-lapse images. The number of pixels inside the well was 16077.98 ± 192.35. The relationship between a pixel and its actual size was 1 pixel = 0.3275 µm.[2,42]

### 4.3. Establishment of the Algorithm for Quantitative Analysis for Pronuclear Assessment

#### 4.3.1. Pronuclear Annotation and Automatic Recognition and Labeling

For the accurate measuring of PN edge and area, a pre-processing of TL images, a Laplacian-based method that could confirm the clearest focal plane from the 11 Z-stack images[43,44] was employed (**Figure 5**A) for both training and calculating databases. In the training database, the edge of each structure (pellucid zone, oocyte membrane, and PNs) was labeled using different color lines by two experienced embryologists. Then, the expert experience of deep learning for recognizing perivitelline space and PN was performed by a mask region-convolutional neural network (Mask R-CNN), which allowed to easily estimate PN poses in the same framework.[45] An abandoned mechanism was introduced to automatic PN recognition, and any abnormal images (0, 1, 3, or more than 3PN) in the frame sequence were discarded. For example, an incorrectly recognized 3PN image classified with all other 2PN images was detected 18 h after ICSI images, and so was discarded in subsequent analysis.

#### 4.3.2. Distinguishing Female and Male PNs

Normally, two separate PNs appear at different times and positions inside the perivitelline space. Three main approaches can

**2200001 (8 of 12)**

**A**

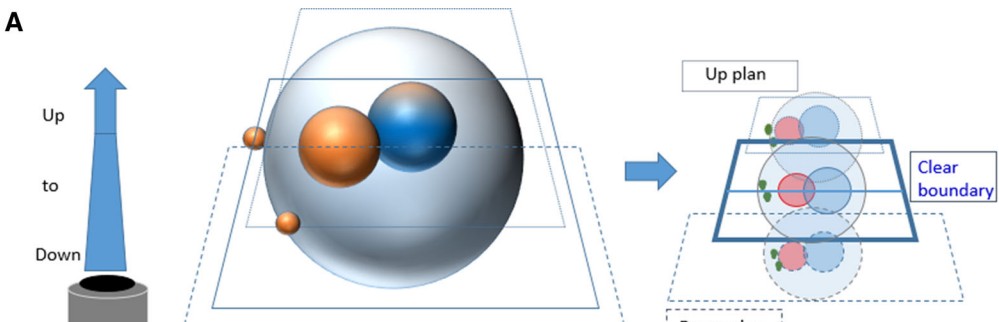

Up
to
Down

Lens

Up plan

Clear boundary

Down plan

**B**

Represent the three positions of male PN ♂ (pronucleus) before the both two PN appearance.

● The male pronuclear centroids

● The female pronuclear centroid

→ The distances of the centroid (Cpm) belong to three preexisting male PN ♂ to the visible centroid of male PN

→ The distances of the centroid (Cpm) belong to three preexisting male PN ♂ to the visible centroid of female PN

dM

dF

**C** a

cosθ

b    Y: area of PN

— male PN
— female PN

time

**D**

Male PN appearance    Female PN appearance    2 PN faint before first mitosis    PN increasing in area during migration

Dynamic area of female PN

Y: PN area/ pixel²

$f(x) = \beta 1 \times \beta 2^{x}, R^{2} \geq 95\%$

$(0, \beta 1)$

Time of PN appearance

Fictitious area of PN at time of ICSI $(0, \beta 1) = \beta 1$

X: Time/hours

be employed for PN classification. First, the position of two separate PNs can be considered. The female PN was closer to the second polar body (PB) than the male PN. Second, the male PN appears earlier than the female PN, but the sequence of pronuclear appearance was hard to differentiate sometimes due to the image quality. Third, male PNs were larger than female PNs in the early zygote stage.[46] However, due to the inherent limitations in automatic labeling, potential inaccurate labeling will be ignored in data outputting. The simpler the annotation, the higher the efficacy that might be obtained in practice. Thus, the PB was not employed as a feature for machine learning but for following handle checking and correction (Figure 2C, type and proportion). Only the second and third methods were employed to distinguish PNs and for automatic PN identification, and the second method was employed before the third method for automatic PN identification (Figure 5B). All pronuclear identifications by computer were confirmed and corrected by a senior embryologist, who did not know the PGT results.

### 4.3.3. The Pronuclear Labeling Stability

In practice, after the orientation of female and male PNs, AI might produce errors when separately labeling and categorizing female or male PNs, so vector calculus discrimination through cosine similarity ($\cos\theta > 0$) was performed for PN location insurance in subsequence images (Figure 5C-a).

### 4.3.4. Exponential Model for Coefficient Extraction

Mathematical models were employed to describe the dynamic nature of pronuclear development, including linear, logarithmic, cosine, quadratic, and exponential functions. Finally, an exponential fitting equation (Figure 5C-b) $f(x) = \beta1 \times \beta2^x$ was employed, and the key coefficient $\beta1$ was extracted from the model. The fitting degree (coefficient of correlation, $R^2$) in the exponential model ranged from 98% to 99.99%, which means that the increase in PN area was fully consistent with the exponential model predictions(for other models of linear, logarithmic, cosine, and quadratic functions, all the $R^2$ value was <90%).

### 4.3.5. Explanation of Coefficients in Mathematical Models

The high value of $R^2$ implies that the development of the PN (from appearance to disappearance) complies with the exponential mathematical model. $\beta1$ represents the fictitious area of the PN at the time of ICSI ($f(0) = \beta1 \times \beta2^0$, where ($f(time of ICSI, t = 0) = \beta1 \times 1$ and $\beta2^0 = 1$) and $\beta2$ represents the PN development

trend (Figure 5D). From this model, any value of the PN area (from 6 to 22 h after ICSI) could be obtained. However, because $\beta2$ was ≈1 (original $\beta2$ mean ± SD: 1.04 ± 0.017, range 1.01 to 1.11), the object of the study was $\beta1$.

### 4.3.6. NPBs Consideration

PN scoring was the assessment of the NPBs that were formed on the chromosomes that contain the DNA for the ribosomal genes and were involved in the formation of the nucleolus at later stages. A recent study reported that zygotes clustering nucleoli at the pronuclear interface were thought to be more likely to develop into healthy euploid embryos.[37] But it was impossible to include NPBs in the model due to current image quality, as well as the continuous alterations in location, size, and movement of NPBs.

### 4.4. Chromosomal Detection in Blastocyst-Stage Embryos

On day 5 (120 h after ICSI), embryos with visible blastocoele were considered blastocysts without considering the quality.

Transferable blastocysts were defined as follows: at least in the blastocyst stage at day 5 with moderate expansion, having easily discernible tightly compacted ICM and having TE either in many cells forming a cohesive epithelium or in few cells forming a loose epithelium.

Several TE cells were extracted for biopsy using mechanical blunt dissection.[47] Following the biopsy, the cells were placed into 0.2-mL thinly walled tubes, which were sealed and frozen by placing them in a freezer at −20 °C before genetic screening. Single-cell, whole-genome amplification (WGA) with multiple annealing and looping-based amplification cycles (MALBAC) was used.

WGA was performed on cleavage-stage blastomeres using MALBAC following the manufacturer's protocol (Catalog No. YK001B; Yikon Genomics).

Cells were lysed by heating (20 min at 50 °C and 10 min at 80 °C) in 5 µL of lysis buffer. Then, 30 µL of the freshly prepared pre-amplification mix was added to each tube and the mixture was incubated at 94 °C for 3 min. Next, DNA was amplified using 8 cycles of 40 s at 20 °C, 40 s at 30 °C, 30 s at 40 °C, 30 s at 50 °C, 30 s at 60 °C, 4 min at 70 °C, 20 s at 95 °C, and 10 s at 58 °C and immediately placed on ice. 30 µL of the amplification reaction mix was then added to each tube and the mixture was incubated at 94 °C for 30 s, followed by 17 cycles of 20 s at 94 °C, 30 s at 58 °C, and 3 min at 72 °C.

Low-coverage (0.3×), genome-sequenced MALBAC products were purified using a DNA purification kit to construct the DNA library.

**Figure 5.** Female pronuclei reveal oocyte or embryo abnormality in the fourth figure. *A total of 16 634 images from 276 embryos of 83 couples. A) Pronuclear annotation and automatic recognition and labeling by AI. B) The rules to distinguish pronuclei. C) The pronuclear labeling stability. a) Pronuclear formation until nuclear envelope breakdown, showing the entire process of pronuclear migration and envelope enlargement in the human zygote. Blue represents the male pronucleus (PN) and red represents the female PN. $\cos\theta$ is the residual spin value between the lines through the center of the PN circle in each consecutive frame. $\cos\theta > 0$ is mandatory to ensure PN location. b) The difference in area between male and female PNs will exist from PN appearance until disappearance. This means that larger or smaller PNs will consistently exist and that the PN's sex could be determined in the early zygote stage. The curves of both male and female PNs were established first, and female and male curves were distinguished later. D) Explanation of coefficients in the models. After distinguishing female from male PNs, parameters of female PNs were extracted by the software ($\beta1$ and $\beta2$). $\beta1$ represents the fictitious area of the PN at the time of ICSI and $\beta2$ represents PN development.

## 4.5. Chromosomal Error Definition

"Chromosome-normal" was defined as follows: subsequently developed embryos (blastocysts) were euploid, without genomic disorders of deletions and/or duplications (including microdeletions and/or -duplications), that is, 46 XN.

"Sole mosaic" was defined as follows: subsequently developed embryos were partial cells with normal chromosomes and the others with abnormal chromosomes either with aneuploidy or without genomic disorders of deletions and/or duplications, that is, 46XN, + mosaic (22) (33%) or 46 XN, dup (16) (p13.3p13.13) (5.7 Mb) (mos, 50%).

Because gametes' chromosomal abnormalities should not be the source of "sole mosaic," embryos with sole mosaic forms were considered mitotic chromosomal separation errors.[35] Thus, in this study, "chromosome-normal" and "sole mosaic" were included in the same group in the results.

"Sole aneuploidy" was defined as follows: subsequently developed embryos were aneuploid without any other errors, that is, 47, XN, +22(×3).

"Sole deletion and/or duplication" indicates that embryos possess small (>10 Mb) or submicroscopic genomic deletions and/or duplications (1 kb to 10 Mb) without mosaic forms, for example, 46, XN, dup (16) (p13.3p13.13) (5.7 Mb).

"Aneuploidy with errors" includes aneuploidy with any other solely chromosomal errors, that is, 47, XN, +22(×3), dup (16) (p13.3p13.13) (5.7 Mb) or 47, XN, +22(×3), +mosaic (22) (33%).

"Euploidy with errors" includes "sole deletion and/or duplication" euploidy with any mosaic forms, that is, 46, XN, dup (16) (p13.3p13.13) (5.7 Mb), +mosaic (22) (33%).

"Complex chromosomal errors" indicates aneuploidy with chromosomal deletion and/or duplication and mosaic forms, for example, 45, XN, (−21), +4q (q12q31.1, ≈89 Mb, ×3), 9p (p20p21.1, ≈32 Mb, ×1, mos, ≈50%).

The other classification of chromosomal errors was explored based on the coincidence of embryos' and patients' (the prominent indication or propositus) chromosomal/genomic abnormalities in PGT-SR. The results were grouped as "embryo's chromosomal error coincident with female," "embryo's chromosomal error coincident with male," "embryo's chromosomal error inconsistent with female," "embryo's chromosomal error inconsistent with male," "sole mosaic embryo," and "chromosomal normal embryo." Coincident errors mean the embryos' chromosomes had complete or partial errors like female or male somatic chromosomes, that is, 46, XX, t(1,16)(q42:q12) in somatic cells and 46, XN, +1q (q42.12→qter, ≈23.9 м, ×3), −16q (q12.1→q24.3, ≈39 м, ×1) in the embryo.

### 4.6. Machine Learning Programming and Statistical Analysis

PN machine learning (83 patients [276 blastocysts]), distinguishing female and male PNs, pronuclear labeling stability certainty, PN ranking order, and automatic mathematical model establishment were performed using Python 3.9.7 (downloaded from https://www.python.org/).

Continuous variables were expressed as mean with standard deviation (SD) and categorical data were expressed as rates. Outliers were considered as missing values. The heterogeneity test for continuous variables, the Chi-square test for trend comparison, Spearman correlation analysis, multiple regression, and multiple logistic regression for relationships were used as appropriate (155 patients [790 blastocysts]). $P < 0.05$ was considered to indicate statistical significance. All analyses were conducted using Stata version 15.1 (StataCorp LLC, College Station, TX, USA).

## 5. Conclusions

We believe that this is the first report on automatic calculation and morphologic quantitative data extraction using expert experience in deep learning based on pronuclear characteristics in human embryos. It helps to advance the non-invasively evaluation of embryos. Further prospective, well-designed studies are needed to validate the availability of quantitative PN assessment in clinical practice.

## Supporting Information

Supporting Information is available from the Wiley Online Library or from the author.

## Acknowledgements

The authors declare that they received no funding in support for this research.

## Conflict of Interest

The authors declare no conflict of interest.

## Author Contributions

J.Y. and Y.W. contributed equally to this work. C.L., W.H., G.H., and X.Z. contributed to conceptualization. J.Y., W.H., G.H., and X.Z. contributed to methodology. Y.W., C.L., and S.X. contributed to data curation. J.Y., Y.W., W.L., Q.Z., K.T., and X.Z. contributed to formal analysis. J.Y., C.L., W.H., and X.Z. contributed to visualization. W.L., S.X., and Q.Z. handled supervision. J.Y., Y.W., and X.Z. wrote the original draft. G.H. and X.Z. reviewed and edited the writing.

## Data Availability Statement

Data and code supporting this work are openly available at GitHub, including a workflow as a Jupyter notebook (https://github.com/hechaohong/nucleu_areas/blob/main/main.ipynb) and anonymized images of embryo development (https://github.com/hechaohong/nucleu_areas). Additional imaging data and information on embryo donors cannot be shared for ethical and legal reasons.

## Peer Review

The peer review history for this article is available in the Supporting Information for this article.

## Keywords

artificial intelligence, expert experience deep learning, mathematical models, pre-implantation genetic tests, pronuclei identification

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
