## [**Supplementary Information**: Record of Transparent Peer Review · Advanced Genetics]

Variation of female pronucleus reveals oocyte or embryo chromosomal copy number variations

Jingwei Yang, Yikang Wang, Chong Li, Wei Han, Weiwei Liu, Shun Xiong, Qi Zhang, Keya Tong, Guoning Huang, Xiaodong Zhang

Date submitted: 12 January 2022

Editors: Kerstin Brachhold, Andrew L. Hufton

1st Peer Review Decision

25 May 2022

Dear Dr Zhang,

Thank you for submitting your manuscript entitled "Variation of female pronucleus reveal oocyte or embryo chromosomal copy number variations: a computer assisted analysis" (Research Article, No. ggn2.202200001) to Advanced Genetics. The reviewer report and comments are included at the end of this e-mail.

I am pleased to inform you that your manuscript has been recommended for publication pending satisfactory revisions. I invite you to respond to the reviewer comments and make the necessary changes to your manuscript.

My apologies that the review process has taken so long. Unfortunately, I had an extremely difficult time in finding reviewers.

Statistics: For original research, please check that your manuscript includes a sub-section entitled "Statistical Analysis" at the end of the Experimental Section that fully describes the following information: 1. Pre-processing of data (e.g., transformation, normalization, evaluation of outliers), 2. Data presentation (e.g., mean \pm SD), 3. Sample size (n) for each statistical analysis, 4. Statistical methods used to assess significant differences with sufficient details (e.g., name of the statistical test including one- or two-sided testing, testing level (i.e., alpha value, P value), if applicable post-hoc test or any alpha adjustment, validity of any assumptions made for the chosen test), 5. Software used for statistical analysis.

Figure legends: Please make sure that all relevant figure legends contain the information on sample size (n), probability (P) value, the specific statistical test for each experiment, data presentation and the meaning of the significance symbol.

A more detailed checklist can be found here: <https://www.advancedsciencenews.com/road-better-data-presentation-dos-donts/>

To submit your revision, go to <https://www.editorialmanager.com/advgenet/> and log in as an Author using your username (*****) and password. Your submission can be found under the menu item "Submissions Needing Revision". The changes to your manuscript should be highlighted in a different color in the primary "Revised Manuscript" file.

Please provide a point-by-point response letter addressed to the reviewers, including a list of changes made and a rebuttal to any comments with which you disagree. You may copy the letter into the "Respond to Reviewers" box (if it is plain text only) or upload it as a "Response Letter to Reviewers" item (if it contains figures, tables, or special formatting such as formulas or equations). If necessary, you may also upload a separate revision cover letter addressed to the editor with any other information not intended for reviewers as a "Cover Letter to Editor" item. You will also be asked to upload a .zip archive containing the production data that will be used if your manuscript is accepted. See below for more details.

We should receive your revised manuscript by 08 Jul 2022. Once we receive your revised manuscript, we will provide a final decision as soon as possible.

We recognize that authors are doing their best to revise manuscripts under challenging circumstances due to the COVID-19 pandemic. Should you need extra time, do not hesitate to contact the editorial office.

Yours sincerely,

Kerstin Brachhold

P.S. Please help avoid delays by referring to the Manuscript Preparation Checklist (<http://www.advgenet.com/authorguidelines>) and use the appropriate article template when preparing your revised manuscript. Please also follow the instructions to prepare and upload your Production Data materials. These include: the full, non-highlighted text of your manuscript (with all figures with a resolution of at least 300 dpi, schemes, and tables) in editable format - Word DOC/X or LaTeX; a short summary (50-60 words) and an eye-catching color image for the Journal's Table of Contents; and, if applicable, Supporting Information* . .

*The Supporting Information document(s) will be published alongside your article and should be non-highlighted and ready for publication. Video formats may also be included.

REVIEWER REPORT:

Please note that reviewers may not be numbered consecutively. Where reviewers have provided additional files, these are available here: *****

EVALUATION:

Reviewer's Responses to Questions

Please rate the importance compared to published work in this subject area.

Reviewer #2: High - Top 15% in the subject area

Reviewer #3: High - Top 15% in the subject area

Please rate the novelty compared to published work in this subject area.

Reviewer #2: Outstanding - Top 5% in the subject area

Reviewer #3: High - Top 15% in the subject area

Which aspects of scholarly presentation require improvement (if any)?

Reviewer #2:

*Clarity

*Language

*Writing style

*Manuscript structure

Reviewer #3:

*Display items

Do the methods, data and analysis (including statistical analysis where applicable) adequately test the hypothesis and support the conclusions?

Reviewer #2: Mostly

Reviewer #3: Mostly

Are the methods, data and analysis described in sufficient detail to be reproduced?

Reviewer #2: Yes

Reviewer #3: Yes

What do you anticipate your overall rating (a mean of importance, novelty and scholarly presentation) would be if the requested revisions are adequately addressed?

Reviewer #2: High - Top 15% in the subject area

Reviewer #3: High - Top 15% in the subject area

COMMENTS TO AUTHOR:

Reviewer #2: This is a very interesting and novel manuscript. However, it is very difficult to understand the grammar that is used; moreover, it appears that the word selection may introduce philosophical or factual errors. For example, in many cases, there is a wording that suggests that is anthromorphic (implies behavior that is "thought out" by an embryo or oocyte). This is clear in the first sentence of the abstract: The pronuclei, existing 16-22 hours after fertilization, appears to have the ability to distinguish good and bad oocytes or embryos due to the unique glance of female and male chromosomes.

This problem is evident throughout the manuscript making extensive editing necessary. In addition, there is a need for a flowchart (similar to Figure S4) that details the work and is presented right up front prior to description of results.

Overall, the results are very interesting and novel. The presentation is not good and implies anthromorphic behavior and more importantly, likely skews the result presentation (so that observations become determinative).

Reviewer #3: 1.They reported a quantitative pronuclear measurement method by applying expert experience deep learning from large annotated datasets. After mathematic reconstruction of pronuclei, significant differences were obtained in chromosome-normal rate and chromosomal small errors such as copy number various by comparing the size of reconstructive female pronucleus. The research method is novel and innovative.

- 2.This study is a quantitative measurement method and cannot be used for qualitative diagnosis.
3.Fig.2 There are spelling errors in the vertical title of the picture.

--

Dr. Kerstin Brachhold, Deputy Editor
Advanced Genetics
E-mail: AdvGenet@wiley.com

<http://www.advgenet.com>

Authors' Response to 1 st Review

03 July 2022

Dear reviewers:

Thank you for the suggestions for our previous manuscript. We have carefully revised the manuscript in line with these comments and we attach herewith a point-by-point explanation of how we have addressed the comments.

Reviewer 2:

Question 1 : The manuscript contains important information and is potentially high quality. The writing is difficult and is not just a language challenge. The writing suggests a lack of clarity regarding the important outcomes or points to stress.

Response: Thank you for your kind comments. We changed the abstract to your request: The characteristics of the human pronuclei (PNs), which exist 16–22 hours after fertilization, appear to serve as good indicators to evaluate the quality of human oocyte and embryo, and may reflect the status of female and male chromosome composition. Here, we report a quantitative PN measurement method that was generated by applying expert experience deep learning from large annotated datasets. After mathematic reconstruction of PNs, significant differences were obtained in chromosome-normal rate and chromosomal small errors such as copy number variants by comparing the size of the reconstructive female PN. After integrating the whole procedure of PN dynamics and adjusting for errors that occurred during PN identification, the results were robust. Notably, all positive prediction results were obtained from the female propositus population. Thus, the size of female PNs may be an important biological factor that mirrors the internal quality of the chromosomal integrity of the oocyte. Embryos that develop from zygotes with larger female PNs may have a reduced risk of copy number variations.

Question 2: Please fix this sentence. "A spermatozoon penetrates into the oocyte, causing a series of events that can be observed through a microscope, such as the cortical granule reaction, which preventpoly-fertilization, expel the second polar body, and promote the formation and migration of two separate pronuclei that contain maternal and paternal chromosomes, respectively."

Response: Thank you for your suggestion, we fixed the sentence to your request: A sperm binds to the oocyte and delivers its chromosomal DNA, triggering the beginning of a series of microscopic events in the zygote including the cortical granule reaction, which prevents poly-fertilization, expelling of the second polar body, and the formation and migration of two separate pronuclei that carry the maternal and paternal chromosomes.

Question 3 : Please check the use of the word "mirror" throughout. It is used in an awkward way. For example, Page 3, line 30: "...the pronuclear stage could be the only way to mirror the internal quality....." This should read: "... characteristics of the pronuclear stage may mirror the internal chromosomal integrity of the oocyte and the sperm."

Response: Thank you for your kind comments. We corrected the usage of "mirror" : In theory, characteristics of the pronuclear stage may mirror the internal chromosomal integrity of the oocyte and the sperm.

Question 4: I don't believe you need to use the term "spermatozoon" throughout. Sperm should be sufficient.

Response: Thank you for your correction. We have fixed "spermatozoon" to "sperm".

Question 5: Figure 2 legend needs to be moved to the same page as the figure itself. There is a Figure 2 legend on page 4 and on page 7.

Response: Thank you for your suggestion. The images are too large that we separate them for clarity. After we combined them, we found that the clarity would be greatly reduced and we are afraid that it would cause you inconvenience during the review process, so we kept the current images.

Question 6: Lines 34-44 provide the conclusions. These are not able to be understood as written.

Response: Thank you for your kind comments. We changed the conclusions to your request: "We believe that this is the first report on automatic calculation and morphologic quantitative data extraction using expert experience in deep learning based on pronuclear characteristics in human embryos. It helps to advance the non-invasively evaluation of embryos. Further prospective, well-designed studies are needed to validate the availability of quantitative PN assessment in clinical practice."

Question 6: The references are completely annoying. They are presented in random format, with random abbreviations for the journals, random punctuation and style.

Response: We apologize for the errors in the formatting of the references, and we have carefully revised them following the journal's requirements. We are sorry for this mistake.

Reviewer 4:

Question 1: I would, however, comment that the revised manuscript is still difficult to understand due to

word selection and grammar issues, even though this was reported in a prior round of review.

Response: Thank you for your kind comments. We made extensive editing the article sentence by sentence, corrected the wrong grammar and vocabulary issues, and made the content expressed more accurate. Some necessary corrections, supplements, and deletions have been made, and all the altered passages have been highlighted in red.

Question 2: I have some concerns about this manuscript with respect to the level of description of the machine-learning workflow and other mathematical modelling, which is in my opinion, not adequately described in the materials and methods section.

Response: Thank you for your correction. We have carefully revised our manuscript according to your suggestions.

For example: Then, the expert experience deep learning for recognizing perivitelline space and PN was performed by a mask region-convolutional neural network (Mask R-CNN), which allowed us to easily estimate PN poses in the same framework 47. An abandoned mechanism was introduced to automatic pronucleus recognition, and any abnormal images (0, 1, 3, or more than 3PN) in the frame sequence were discarded. For example, an incorrectly recognized 3PN image classified with all other 2PN images was detected 18 hours after ICSI images, and so was discarded in subsequent analysis.

For example: Mathematical models were employed to describe the dynamic nature of pronuclear development, including linear, logarithmic, cosine, quadratic, and exponential functions. Finally, an exponential fitting equation (Fig. 5C-b) β_2x was employed, and the key coefficient β_1 was extracted from the model. The fitting degree (coefficient of correlation, R²) in the exponential model ranged from 98% to 99.99%, which means that the increase in PN area is fully consistent with the exponential model predictions (for other models of linear, logarithmic, cosine, and quadratic functions, all the R² value is <90%).

Question 3: Overall, I feel the work would have much greater impact if the authors provided their code/models for peer-review and reuse. I would also suggest the authors at a minimum provide their code (e.g. via github) but optimally, provide this as a reusable workflow (e.g. in a Jupyter notebook) that would fully describe the steps taken in model creation and validation. If possible, I would encourage the authors to also publish the training/validation datasets (anonymized, of course) such that the work could be reproduced, properly evaluated, and compared to, for example, models created by others.

Response: We provide our code (via github:

https://github.com/hechaohong/nucleu_areas/blob/main/main.ipynb) with a workflow (Jupyter notebook) and images of an embryo development (via github:

https://github.com/hechaohong/nucleu_areas). We are sorry that for legal reasons we can only provide data related to an anonymized embryo. Thank you for your suggestion !

Dear Dr Zhang,

Thank you for submitting your revised manuscript entitled "Variation of female pronucleus reveal oocyte or embryo chromosomal copy number variations: a computer-assisted analysis" (Research Article, No. ggn2.202200001R1) to Advanced Genetics. The reviewers' reports and comments are included at the end of this e-mail.

As you will see, the reviewers felt that the manuscript had been improved during revision, but felt that additional work was needed to ensure that the manuscript meets the journal's high standards for clarity and FAIR data and code sharing.

Both reviewers felt that the manuscript would benefit from some editing to improve the grammar and clarity of the text. We would encourage you to work with a native English speaker or an appropriate science editing service to help improve the manuscript.

Advanced Genetics is committed to publishing science that is as FAIR and open as possible. Please therefore consider in detail the comments from Reviewer #4 and specifically their final comment: "I would also suggest the authors at a minimum provide their code (e.g. via github) but optimally, provide this as a reusable workflow (e.g. in a Jupyter notebook) that would fully describe the steps taken in model creation and validation. If possible, I would encourage the authors to also publish the training/validation datasets (anonymized, of course) such that the work could be reproduced, properly evaluated, and compared to, for example, models created by others."

Data and code files can be provided through an appropriate DOI-issuing scientific repository such as figshare (<https://figshare.com/>), Zenodo (<https://zenodo.org/>) or OSF (<https://osf.io/>). All of these repositories offer integration with github, if you want to host code files there as recommended by Reviewer #4. Your final data record should be mentioned in your data availability statement and listed in the main References section using the citation information recommended by your chosen data repository. Please let us know if addressing this request would be a problem for any reason.

Regarding the comment from Reviewer #2 about the References section, please note that this section will be thoroughly edited and re-formatted by the journal's copy-editors if this work is acceptable for publication. We would encourage you to standardize the journal names somewhat to assist Reviewer #2, but perfect formatting is not needed at this stage.

If these final issues can be fully addressed, we are hopeful that your next revised manuscript version may be found acceptable for publication at the journal. Every effort will be made to re-evaluate your next version as quickly as possible. I therefore invite you to respond to the reviewer comments and make the necessary changes to your manuscript.

Statistics: For original research, please check that your manuscript includes a sub-section entitled "Statistical Analysis" at the end of the Experimental Section that fully describes the following information: 1. Pre-processing of data (e.g., transformation, normalization, evaluation of outliers), 2.

Data presentation (e.g., mean \pm SD), 3. Sample size (n) for each statistical analysis, 4. Statistical methods used to assess significant differences with sufficient details (e.g., name of the statistical test including one- or two-sided testing, testing level (i.e., alpha value, P value), if applicable post-hoc test or any alpha adjustment, validity of any assumptions made for the chosen test), 5. Software used for statistical analysis.

Figure legends: Please make sure that all relevant figure legends contain the information on sample size (n), probability (P) value, the specific statistical test for each experiment, data presentation and the meaning of the significance symbol.

A more detailed checklist can be found here: <https://www.advancedsciencenews.com/road-better-data-presentation-dos-donts/>

To submit your revision, go to <https://www.editorialmanager.com/advgenet/> and log in as an Author using your username (*****) and password. Your submission can be found under the menu item "Submissions Needing Revision". The changes to your manuscript should be highlighted in a different color in the primary "Revised Manuscript" file.

Please provide a point-by-point response letter addressed to the reviewers, including a list of changes made and a rebuttal to any comments with which you disagree. You may copy the letter into the "Respond to Reviewers" box (if it is plain text only) or upload it as a "Response Letter to Reviewers" item (if it contains figures, tables, or special formatting such as formulas or equations). If necessary, you may also upload a separate revision cover letter addressed to the editor with any other information not intended for reviewers as a "Cover Letter to Editor" item. You will also be asked to upload a .zip archive containing the production data that will be used if your manuscript is accepted. See below for more details.

We should receive your revised manuscript by 21 Aug 2022. Once we receive your revised manuscript, we will provide a final decision as soon as possible.

We recognize that authors are doing their best to revise manuscripts under challenging circumstances due to the COVID-19 pandemic. Should you need extra time, do not hesitate to contact the editorial office.

Yours sincerely,

Andrew Hufton

P.S. Please help avoid delays by referring to the Manuscript Preparation Checklist (<http://www.advgenet.com/authorguidelines>) and use the appropriate article template when preparing your revised manuscript. Please also follow the instructions to prepare and upload your Production Data materials. These include: the full, non-highlighted text of your manuscript (with all figures with a resolution of at least 300 dpi, schemes, and tables) in editable format - Word DOC/X or LaTeX; a short summary (50-60 words) and an eye-catching color image for the Journal's Table of Contents; and, if applicable, Supporting Information*.

*The Supporting Information document(s) will be published alongside your article and should be non-highlighted and ready for publication. Video formats may also be included.

REVIEWER REPORT:

Please note that reviewers may not be numbered consecutively. Where reviewers have provided additional files, these are available here: *****

EVALUATION:

Reviewer's Responses to Questions

Please rate the importance compared to published work in this subject area.

Reviewer #2: Outstanding - Top 5% in the subject area

Reviewer #4: (No Response)

Please rate the novelty compared to published work in this subject area.

Reviewer #2: High - Top 15% in the subject area

Reviewer #4: (No Response)

Which aspects of scholarly presentation require improvement (if any)?

Reviewer #2:

*Clarity

*Language

*Writing style

*Manuscript structure

*References

Reviewer #4:

- *Clarity
- *Language
- *Writing style
- *Display items
- *Supporting Information

Do the methods, data and analysis (including statistical analysis where applicable) adequately test the hypothesis and support the conclusions?

Reviewer #2: Mostly

Reviewer #4: Partially

Are the methods, data and analysis described in sufficient detail to be reproduced?

Reviewer #2: Yes

Reviewer #4: No

Where applicable, have the requested revisions been adequately addressed?

Reviewer #2: Mostly

Reviewer #4: Partially

COMMENTS TO AUTHOR:

Reviewer #2: The manuscript contains important information and is potentially high quality. The writing is difficult and is not just a language challenge. The writing suggests a lack of clarity regarding the important outcomes or points to stress.

Point #1: Please clarify the abstract. Suggested wording is below.

Current abstract:

Abstract

The pronucleus (PN), which exists 16-22 hours after fertilization, appears to serve as a good indicator of

distinguishing oocyte or embryo quality due to presenting the unique appearance of female and male chromosomes. Here, we reported a quantitative PN measurement method by applying expert experience deep learning from large annotated datasets. After mathematic reconstruction of PNs, significant differences were obtained in chromosome-normal rate and chromosomal small errors such as copy number various by comparing the size of the reconstructive female PN. The results were robust after integrating the whole procedure of PN dynamics and adjusting the errors occurred during PN identification. Notably, all positively predictive results were obtained from the female propositus population. Thus, female PN size, an essential biological factor, could mirror the internal quality of the chromosomal integrity of the oocyte. The embryos developing from a big female PN zygote could have a lower possibility of copy number variations compared with those developed from a small PN zygote.

Change to:

Abstract

Characteristics of the human pronuclei (PNs), which exist 16-22 hours after fertilization, appear to serve as good indicators to distinguish human oocyte and embryo quality and may reflect status of female and male chromosome composition. Here, we report a quantitative PN measurement method that was generated by applying expert experience deep learning from large annotated datasets. After mathematic reconstruction of PNs, significant differences were obtained in chromosome-normal rate and chromosomal small errors such as copy number variants by comparing the size of the reconstructive female PN. After integrating the whole procedure of PN dynamics and adjusting for errors that occurred during PN identification, the results were robust. Notably, all positive predictive results were obtained from the female propositus population. Thus, female PN size may be an important biological factor that mirrors the internal quality of the chromosomal integrity of the oocyte. Embryos that develop from a zygote with a larger female PN may have a reduced risk of copy number variations.

Point #2: Please fix this sentence.

"A spermatozoon penetrates into the oocyte, causing a series of events that can be observed through a microscope, such as the cortical granule reaction, which preventpoly-fertilization, expel the second polar body, and promote the formation and migration of two separate pronuclei that contain maternal and paternal chromosomes, respectively."

Change to:

"A spermatozoon binds to the oocyte and delivers its' chromosomal DNA triggering the beginning of a series of microscopic events in the zygote including the cortical granule reaction, which prevents poly-fertilization, expelling of the second polar body, and the formation and migration of two separate pronuclei that carry the maternal and paternal chromosomes."

Point #3: Please check the use of the word "mirror" throughout. It is used in an awkward way. For example, Page 3, line 30: "...the pronuclear stage could be the only way to mirror the internal quality....." This should read: "... characteristics of the pronuclear stage may mirror the internal chromosomal integrity of the oocyte and the sperm."

Point #4: I don't believe you need to use the term "spermatozoon" throughout. Sperm should be sufficient.

Point #5: Page 3, line 58 should read "indistinguishable" rather than "undistinguishable."

Point #6: Figure 2 legend needs to be moved to the same page as the figure itself. There is a Figure 2 legend on page 4 and on page 7.

Page 8: In Lines 10-11, it should read as follows: "The baseline characteristics of patients and their IVF outcomes is shown in Table 1."

Page 28: Lines 34-44 provide the conclusions. These are not able to be understood as written. I believe that the authors are suggesting: 1) This is the first report of automatic calculation and morphologic quantitative data extraction via use of expert experience deep learning in human embryos. I would probably modify that a bit as there are other AI type reports. It should read: We believe that this is the first report of automatic calculation and morphologic quantitative data extraction via use of expert experience deep learning based on pronuclear characteristics in human embryos. 2) The sentence that begins with "The results suggest Is simply not able to be understood (with references 39-42). 3) Further high-quality design studies..... is also a sentence that is not clear in meaning.

The conclusions should be written clearly on page 28 if the article is to have any meaningful impact.

Page 29: The references are completely annoying. They are presented in random format, with random abbreviations for the journals, random punctuation and style. This is very sloppy and would normally suggest that the authors have invested minimally in providing a well-written manuscript.

Reviewer #4: The authors present an approach to measuring oocyte quality based on a machine-aided classification that associated positively with oocyte quality.

As a member of the Advanced Genetics editorial board, I have been asked by the primary Editor of this submission to comment primarily with respect to consideration of the reusability and reproducibility of the work, and its adherence to the FAIR (Findable, Accessible, Interoperable, Reusable) data principles. I would, however, comment that the revised manuscript is still difficult to understand due to word selection and grammar issues, even though this was reported in a prior round of review.

With respect to FAIR, Advanced Genetics considers issues of reproducibility to be of significant importance. I have some concerns about this manuscript with respect to the level of description of the machine-learning workflow and other mathematical modelling, which is in my opinion, not adequately described in the materials and methods section. For example:

"Then, the expert experience features training for recognizing perivitelline space and PN was performed by a mask region-convolutional neural network (Mask R-CNN), which allowed us to easily estimate PN poses in the same framework 47. An abandoned mechanism was introduced to automatic pronucleus recognition, and any abnormal images (0, 1, 3, or more than 3PN) were discarded."

The weights provided to the pytorch model builder are not described, and it isn't clear what "an abandoned mechanism" refers to.

Similarly:

"Mathematical models were employed to describe the dynamic nature of pronuclear development, including linear, logarithmic, cosine, quadratic, and exponential functions. Finally, an exponential fitting equation (Fig. 5C-b) $y = \beta_1 \times \beta_2^x$ was employed and the key coefficient β_1 was extracted from the model (for all others, $R^2 < 90\%$)."

This seems to be an inadequate description of the process of model development/evaluation, and the justification of the final selection.

Overall, I feel the work would have much greater impact if the authors provided their code/models for peer-review and reuse. I would also suggest the authors at a minimum provide their code (e.g. via github) but optimally, provide this as a reusable workflow (e.g. in a Jupyter notebook) that would fully describe the steps taken in model creation and validation. If possible, I would encourage the authors to also publish the training/validation datasets (anonymized, of course) such that the work could be reproduced, properly evaluated, and compared to, for example, models created by others.

--

Andrew Lee Hufton, PhD, Editor-in-Chief
Advanced Genetics
E-mail: AdvGenet@wiley.com

<http://www.advgenet.com>

3rd Editor's Decision	13 September 2022
-------------------

Dear Dr Zhang,

Thank you for submitting your revised manuscript entitled "Variation of female pronucleus

reveals oocyte or embryo chromosomal copy number variations: a computer-assisted analysis" (Research Article, No. ggn2.202200001R2) to Advanced Genetics. This work was assessed again by Reviewer #4, who is now fully satisfied with the changes made. We are therefore delighted to accept this work, in principle, for publication, pending your ability to address the minor points below.

1. Please update your data availability statement to describe the code, workflow and sample images you have now provided through github. Github links should be provided in the text of the statement. Please update the data availability statements in both your manuscript and in our submission system.

2. I would recommend that you shorten the title to: "Variation of female pronucleus reveals oocyte or embryo chromosomal copy number variations". Shorter titles tend to attract more readers, and most publications at the journal are "computer-assisted" in one form or another.

To submit your revision, go to <https://www.editorialmanager.com/advgenet/> and log in as an Author using your username (*****) and password. Your submission can be found under the menu item "Submissions Needing Revision". The changes to your manuscript should be highlighted in a different color in the primary "Revised Manuscript" file.

We should receive your revised manuscript by 23 Sep 2022. Once we receive your revised manuscript, we will provide a final decision as soon as possible. Should you need extra time, do not hesitate to contact the editorial office.

Yours sincerely,

Andrew Hufton

P.S. Please help avoid delays by referring to the Manuscript Preparation Checklist (<http://www.advgenet.com/authorguidelines>) and use the appropriate article template when preparing your revised manuscript. Please also follow the instructions to prepare and upload your Production Data materials. These include: the full, non-highlighted text of your manuscript (with all figures with a resolution of at least 300 dpi, schemes, and tables) in editable format - Word DOC/X or LaTeX; a short summary (50-60 words) and an eye-catching color image for the Journal's Table of Contents; and, if applicable, Supporting Information* . .

*The Supporting Information document(s) will be published alongside your article and should be non-highlighted and ready for publication. Video formats may also be included.

--

Dr Andrew Hufton, Editor

Advanced Genetics
E-mail: AdvGenet@wiley.com

<http://www.advgenet.com>

Final Decision	26 September 2022
----------------	-------------------

Dear Dr Zhang,

Thank you for submitting your manuscript entitled "Variation of female pronucleus reveals oocyte or embryo chromosomal copy number variations" (Research Article, No. ggn2.202200001R3) to Advanced Genetics. I'm pleased to inform you that your manuscript has now been accepted for publication.

To confirm our recent correspondence, your data availability statement will be updated as follows during final production:

"Data and code supporting this work are openly available at GitHub, including a workflow as a Jupyter notebook (https://github.com/hechaohong/nucleu_areas/blob/main/main.ipynb) and anonymized images of embryo development (https://github.com/hechaohong/nucleu_areas). Additional imaging data and information on embryo donors cannot be shared for ethical and legal reasons."

We will copyedit the accepted version of your manuscript and if we require any further information at this stage we will contact you. After copyediting we will let you know when you can expect to receive the proofs. Instructions for returning your proof corrections will be provided when the proofs are sent to you.

All articles published in Advanced Genetics are fully open access: immediately and freely available to read, download and share. Advanced Genetics charges a publication fee to cover publication costs. The corresponding author for this manuscript should have already received a quote with the article publication fee, and will soon receive an e-mail invitation to register with or log in to Wiley Author Services (<https://authorservices.wiley.com>). After logging into Wiley Author Services, the publication fee can be paid by credit card, or an invoice or pro forma can be requested. Payment of the publication charge must be received before the article will be published online.

Congratulations on your results, and thank you for choosing Advanced Genetics for publishing your work. I hope you will consider us for the publication of your future manuscripts.

Yours sincerely,

Andrew Hufton

P.S.: If you believe your images might be appropriate for use on the cover of Advanced Genetics, and you would like your paper to be considered for the cover, please e-mail us your layout suggestions with a

short description. For details on cover image preparation, please see the cover gallery on <http://www.advgenet.com>.

--

Dr Andrew Hufton, Editor
Advanced Genetics
E-mail: AdvGenet@wiley.com

<http://www.advgenet.com>